# Valorization of Honduran Agro-Food Waste to Produce Bioplastics

**DOI:** 10.3390/polym15122625

**Published:** 2023-06-09

**Authors:** Daniel Castro-Criado, Octavio Rivera-Flores, Johar Amin Ahmed Abdullah, Elia Castro-Osorto, María Alonso-González, Lucy Ramos-Casco, Víctor M. Perez-Puyana, Marlon Sánchez-Barahona, Pablo Sánchez-Cid, Mercedes Jiménez-Rosado, Alberto Romero

**Affiliations:** 1Departamento de Ingeniería Química, Universidad de Sevilla, 41012 Sevilla, Spain; dancascri@alum.us.es (D.C.-C.); jabdullah@us.es (J.A.A.A.); maralonso@us.es (M.A.-G.); vperez11@us.es (V.M.P.-P.); psanchezcid@us.es (P.S.-C.); mjimenez42@us.es (M.J.-R.); 2Unidad de Gestión de Investigación Científica, Ingeniería Agroindustrial, Universidad Nacional Autónoma de Honduras Tecnológico Danlí, Danlí 13201, Honduras; jorivera@unah.edu.hn (O.R.-F.); ecastro@unah.edu.hn (E.C.-O.); lucy.ramos@unah.edu.hn (L.R.-C.); marlon.sanchez@unah.edu.hn (M.S.-B.)

**Keywords:** bioplastics, taro, yucca, banana, valorization, circular economy

## Abstract

The development of biodegradable plastics and eco-friendly biomaterials derived from renewable resources is crucial for reducing environmental damage. Agro-industrial waste and rejected food can be polymerized into bioplastics, offering a sustainable solution. Bioplastics find use in various industries, including for food, cosmetics, and the biomedical sector. This research investigated the fabrication and characterization of bioplastics using three types of Honduran agro-wastes: taro, yucca, and banana. The agro-wastes were stabilized and characterized (physicochemically and thermically). Taro flour presented the highest protein content (around 4.7%) and banana flour showed the highest moisture content (around 2%). Furthermore, bioplastics were produced and characterized (mechanically and functionally). Banana bioplastics had the best mechanical properties, with a Young’s modulus around 300 MPa, while taro bioplastics had the highest water-uptake capacity (200%). In general, the results showed the potential of these Honduran agro-wastes for producing bioplastics with different characteristics that could add value to these wastes, promoting the circular economy.

## 1. Introduction

In recent years, there has been a significant increase in the use of non-renewable resources and synthetic materials to produce plastics, with fossil fuels being the primary source [1]. The most common plastic types, such as polyethylene, polypropylene, and polyvinyl chloride (PVC), are derived from crude oil. Additionally, plastics like polystyrene, nylon, and acrylics are manufactured using synthetic materials [2]. These materials have garnered interest in multiple industries due to their diverse range of desirable characteristics, such as flexibility, adaptability, and impermeability [3]. Nevertheless, the overreliance on non-renewable resources has resulted in a detrimental impact on the environment, with plastics pollution being a significant concern due to their lack of biodegradability. The overuse of plastics has severe consequences, especially for the oceans, which have become dumping grounds for tons of plastic waste. Therefore, urgent action is needed to promote sustainable alternatives such as biodegradable plastics and eco-friendly biomaterials [4,5]. Promoting a shift towards biodegradable plastics is crucial for reducing the environmental damage caused by non-renewable resources and synthetic materials.

One effective way of achieving this is through the development of eco-friendly biomaterials, such as using renewable resources derived from agro-industrial products, such as vegetable matter [6,7,8]. Agro-wastes, which include discarded plant matter and rejected food, have emerged as potential raw materials for bioplastic production [9]. Their utilization offers a sustainable alternative to traditional plastics derived from fossil fuels. Moreover, harnessing agro-wastes contributes to the concept of the circular economy by promoting resource efficiency and waste reduction [10]. However, there is a need for comprehensive analysis and evaluation of the physicochemical and thermal properties of specific agro-wastes to determine their suitability for bioplastic development [10,11]. Understanding the unique characteristics of agro-wastes is essential for optimizing processing conditions and enhancing the properties of bioplastics. 

These materials are biodegradable and compostable and can be used in various industries. Thus, they could be used in the food industry to prevent food from deteriorating due to factors such as dehydration, light exposure, and oxidation, and they can also accommodate a wide range of additives including antifungals, antimicrobials, antioxidants, flavorings, and nutrients to enhance the quality of food [12,13,14]. They are also used in the cosmetics industry to improve the effectiveness of hair gels, shampoos, and other beauty products while also providing consumers with an eco-friendly choice [15,16,17,18]. They possess inherent properties that render them ideal for biomedical purposes, including tissue engineering, wound care, and gene therapy. Furthermore, they may exhibit a wide range of activities, such as antidiabetic, antihypertensive, and anticancer [19,20,21,22]. Additionally, bioplastics also find use in other sectors, including paints and fertilizers, indicating their versatility and potential as an eco-friendly substitute for conventional plastics [23,24,25]. Several studies have explored the use of agro-wastes such as rice straw [26], winemaking by-products [27], and banana peels [28] for the production of bioplastics, showcasing their potential. In this way, several works promote the use of agro-wastes to conform bioplastics [29]. 

By valorizing agro-wastes as raw materials for bioplastics, we can add value to these materials and promote the circular economy in this sector [23]. In this work, our focus was on three specific Honduran agro-wastes: taro, yucca, and banana, which are commonly used as soil amendments or animal feed. Each of these agro-wastes presents distinct advantages for bioplastic production. Taro, with its high starch content, serves as an excellent source for bioplastic materials and offers cost-effective large-scale production opportunities [30]. Yucca, a tropical crop naturally resistant to pests and diseases, provides a sustainable and environmentally friendly option, with its high yield and ease of cultivation making it a reliable source of raw materials for bioplastics [31]. Banana, known for its high fiber content, contributes additional strength and durability to bioplastics, and its widespread availability and affordability make it practical for commercial-scale production [32]. However, the existing research lacks any comprehensive analysis and evaluation of the physicochemical and thermal properties of these Honduran agro-wastes as raw materials for bioplastics.

The main purpose of this study was to bridge this gap by providing a complete analysis and evaluation of the three different Honduran agro-wastes (taro, yucca, and banana) as raw materials for the development of bioplastics. This study aimed to explore the physicochemical and thermal properties of these agro-wastes and optimize their processing conditions for the development of bioplastics.

In this sense, the main objective of this work was the search for new alternatives in the production of raw materials for the fabrication of bioplastics in a region of Honduras, a developed country. Thus, a complete study from agro-waste to final added-value product was developed. The findings of this research will provide valuable insights into the potential of these agro-wastes as sustainable alternatives to traditional plastics made from non-renewable resources. The study’s outcomes can contribute to the promotion of the circular economy and the development of eco-friendly biomaterials, which have diverse applications in various industries, including for food, cosmetics, biomedical products, paints, and fertilizers.

## 2. Materials and Methods

### 2.1. Materials

The different agro-wastes (taro, yucca, banana) were supplied by Mesa SAN R11 El Paraíso (Danli, El Paraíso, Honduras). These agro-wastes consisted of damaged or deformed fruits that could not be sold on the market due to hygienic or quality problems. In addition, glycerol and distilled water were used as plasticizers for the manufacture of bioplastics, and these were provided by Distribuidora del Caribe S.A. (San Pedro Sula, Honduras).

### 2.2. Raw Material Obtention

The different agro-wastes needed to be treated to stabilize and use them as raw materials for bioplastic production. In this way, they were treated to obtain flour. The flour preparation process is indicated in Figure 1.

Firstly, the three samples were washed with water. Later, they were cut into thin slices (thickness: 0.5 cm approximately) using a laminating machine HLC-300 (Intertek, London, UK). This process was carried out to optimize the drying process by increasing the specific surface area of the material. Then, the samples were submitted to a drying process in an oven at 60 °C for 4 h to prevent denaturation of the polymers and, after that, they were tempered at room temperature for 40 min. This drying process was carried out in order to prevent the high water content of these samples (>50%) from destabilizing the flours obtained during storage. In this way, the elimination of water allows the flours to be stored for a longer time without the appearance of microorganisms or fungi that alter their physicochemical characteristics.

Once dried, the slices were fed into a rotating ball mill LIHONG LHG-1.5 (Agropec, Madrid, Spain) where the friction of the balls breaks the particles and gradually reduces their size until they pass through a 60-micron sieve (250 mesh). This process allows the obtention of the flours.

It is worth mentioning that the samples were weighed after each step of the process to assess the yield of the flour obtained.

### 2.3. Flour Characterization

#### 2.3.1. Chemical Composition

The chemical composition of the flour was determined following the approved methods of A.O.A.C [33]. Moisture content was measured after subjecting the samples to a heat treatment at 105 °C in a JP Selecta oven (Barcelona, Spain) for 24 h. Ash content was obtained after calcining the volatile matter at 550 °C in an HD-240 furnace (Hobersal, Barcelona, Spain) for 5 h. A solid–liquid extraction was performed using the Soxhlet method to obtain the lipid content [34]. Proteins were obtained by elemental analysis in a LECO CHNS-932 nitrogen microanalyzer, applying a correction factor of 6.5 [35]. Finally, polysaccharide content was determined following the protocol used by He et al. [36]. Briefly, 0.5 g of the sample was mixed with 0.5 mL of 5% phenol and 2.5 mL concentrated H_2_SO_4_ in an ice-water bath. Then, it was heated in boiling water for 15 min. The absorbance of the mixture was measured at 490 nm using a spectrophotometer Genesys-20 (Thermo Scientific, Waltham, MA, USA). A calibration curve was performed with sucrose (between 0 and 120 wt% concentration in sucrose with 15 points).

#### 2.3.2. Fourier Transform Infrared Spectroscopy (FTIR)

The molecular composition and intermolecular interactions of the specimens were assessed using Fourier Transform Infrared Spectroscopy (FTIR). The FTIR analysis was performed using a Hyperion 100 spectrometer (Bruker, Billerica, MA, USA) equipped with an Attenuated Total Reflectance (ATR) diamond sensor. Spectral data were collected in the wavenumber range of 4000 to 500 cm^−1^, with a resolution of 4 cm^−1^ and an averaging of 200 scans.

#### 2.3.3. Granulometry

The particle size of the different flours was evaluated according to ISO 13320 [37] using a laser-diffraction particle-size analyzer (Malvern Instrument, Malvern, UK) in a dry process. Briefly, the flour was aspirated by the equipment and passed through lenses where the incident laser was altered by the size of the particles that make up the flour. The change in the angle of the laser was measured and a relationship established to determine the size of the particle (between 0.1 and 1000 µm), making it possible to obtain a size distribution.

#### 2.3.4. Differential Scanning Calorimetry (DSC)

DSC experiments were performed with a Q20 (TA Instruments, New Castle, DE, USA) using 3–8 mg samples, in hermetic aluminum pans. The analyses were performed between 20 and 250 °C at a heating rate of 10 °C/min. The sample was purged with a nitrogen flow of 50 mL/min.

#### 2.3.5. Thermogravimetric Analysis (TGA)

TGA were carried out using simultaneous DSC/TGA equipment, the Q600 (TA Instruments, New Castle, DE, USA), to study the thermomechanical stability of the different flours. In these analyses, the samples were introduced into the equipment at a temperature of 25 °C, and then the temperature was raised to 500 °C at a speed of 10 °C/min. During the test, the samples were kept in a nitrogen atmosphere and their weight variation was measured.

### 2.4. Bioplastics Processing

The bioplastic processing was performed in three different stages where raw materials were mixed, homogenized, and conformed.

Firstly, the materials were mixed using a Hamilton Beach 63227 (Intertek, London, UK) kitchen mixer until a homogeneous dough-like blend was obtained. This process was performed for 22 min in order to obtain a blend that could be incorporated into the extruder machine (the next stage). The selected proportions were 70, 20, and 10 wt.% of flour, glycerol, and distilled water, respectively. This flour:plasticizer ratio (7:3) was chosen based on previous tests where raw materials with similar composition were mixed in this same ratio with plasticizer to preform bioplastics [38]. Water was used due to its beneficial effect in the starch gelatinization process, which requires shear forces, temperature, and the presence of water [39]. Nevertheless, the use of water alone as a plasticizer produces highly fragile products with little flexibility, so the presence of some other plasticizer such as glycerol or sorbitol is recommended. In this case, glycerol was used in a 2:1 (glycerol:water) ratio, in accordance with previous works [40]. 

The blends obtained after the mixing stage were processed through extrusion to produce pellets that could be shaped as bioplastics. The in-house-manufactured extruder (Appendix A) had a screw with a 2.1 cm diameter with a gap between the teeth of 1.5 cm. The screw chamber measured 30 cm with three heating zones (placed equidistantly, each zone measuring 10 cm). Thus, a temperature profile of 60, 80, and 90 °C was selected to perform this stage. These temperatures were chosen considering that starch gelatinization occurs between 70 and 90 °C [41], expecting to destroy the granular structure of native starch with the increase of temperature. Finally, the motor that drives the screw speed was rated at 130 W and its speed was fixed at 85 rpm. The blends were introduced into the feed hopper and, during the first pass, blend conformation was produced, although it was observable to the naked eye that there were heterogenous zones where there was flour that had not been integrated with the plasticizer. For this reason, a second pass was performed. This pass allowed perfect integration of the raw materials as pellets that kept their shape and could be used for injection. This extruder had a 10 mm nozzle to obtain cylindrical products with a constant section.

Finally, pellets were shaped by injection molding to produce bioplastics. The injection equipment consisted of an in-house-manufactured screw injector with four zones where the temperatures to which the samples were subjected could be controlled, in addition to the mold temperature. The temperatures selected were 50, 55, 60, and 100 °C. In this way, the pellets advanced along the screw at between 50 and 60 °C, which was sufficient for them to flow correctly and heat up without negatively affecting the nature of the raw material. Finally, the mold was set at 100 °C, a temperature that allows bioplastic conformation.

### 2.5. Bioplastics Characterization

#### 2.5.1. Dynamic-Mechanical Tests

Dynamic-strain and frequency-sweep tests were performed on a dynamic-mechanical analyzer RSA-3 (TA Instruments, New Castle, DE, USA) in tension and flexural modes. For these, tension and dual-cantilever geometries were used, respectively. Firstly, strain-sweep tests were carried out at a constant frequency of 1 Hz in a strain range of 0.01 to 1% to determine the linear viscoelastic range. Subsequently, frequency-sweep tests were carried out at between 0.2 and 20 Hz at a constant strain (in the linear viscoelastic range). In these tests, the values of elastic (E′) and viscous (E″) moduli were obtained.

#### 2.5.2. Static Tests

Tensile static tests were performed using Insight 10 kN electromechanical testing equipment (MTS, Eden Prairie, MN, USA) according to ISO 527-2:2012 [42]. Stress–strain curves were obtained with a rate of 10 mm/min. In addition, the values of Young’s modulus (E), maximum stress (σ_maz_), and strain at break (ε_max_) for the different systems were calculated to compare the different systems.

#### 2.5.3. Water-Uptake Capacity (WUC)

The water-uptake capacity (*WUC*) values were evaluated following the ASTM D570 standard [43]. For this, the samples were immersed in distilled water and weighed after 24 h to obtain the wet-mass value (wet weight). Finally, they were subjected to a freeze-drying process at −80 °C under vacuum conditions in a LyoQuest freeze-dryer with a Flask M8 head (Telstar, Terrassa, Spain) after being previously frozen at −40 °C. After this drying, the final dry weight was obtained. The value of *WUC* was obtained using the following equation:(1)$$WUC\left(\%\right)=\frac{Wet\;weight-Final\;dry\;weight}{Final\;dry\;weight}\cdot 100$$

### 2.6. Statistical Analysis

At least three replicates of each measure were made. Statistical analyses were carried out with t tests and one-way analysis of variance (*p* < 0.05). For this, Statgraphics (Statgraphics Technologies, The Plains, VA, USA) was used. The standard deviations of the selected parameters were also calculated.

## 3. Results and Discussion

### 3.1. Flour Obtention: Process Performance

The yields of each stage of the flour obtention process are listed in Table 1. It can be seen that during washing and peeling, a small part of the residue was discarded, this being especially notable in the bananas. This was due to the fact that this washing stage also consisted of peeling the fruit, to avoid an excess of fibers in the samples (the peels are mainly composed of this component). For this reason, the bananas had the lowest yield as they have the thickest peels. On the other hand, it is important to emphasize the low yields observed in the drying stage. These low yields were due to the large amount of water contained in the mixtures, as previously observed. Therefore, a low yield means that a lot of water has been extracted from these samples, which is the result that was expected in this stabilization step.

### 3.2. Flour Characterization

#### 3.2.1. Chemical Composition

Table 2 shows the results obtained from the chemical characterization of each system. Firstly, it is worth noting the small amount of moisture found in the samples once stabilized. This small amount of water means that the flours can be kept for a longer time, as previously mentioned. This low water content is related to the low yield of the drying stage during the process, since a large amount of water was eliminated. In addition, the highest percentage in the chemical characterization was associated with starch content, which according to other studies corresponds to amylose and amylopectin, with the latter being the one found in greater proportions [44]. This fraction, together with the protein fraction (which was much smaller), is the one that suggests that a material could have great potential as a raw material for producing bioplastics, since most of this material would be made up of biopolymers. The rest of the unidentified components were assumed to be fibers present in the composition of these fruits. These fibers could serve as filler reinforcing the bioplastics [45].

#### 3.2.2. FTIR

FTIR characterization of the different flours was performed to identify functional groups and absorbance peaks in their FTIR spectra (Figure 2A). The results revealed characteristic peaks corresponding to various functional groups, such as hydroxyl (OH), carbonyl (C=O), and starch (C-O-C) groups, indicating the presence of potential biopolymer candidates for the fabrication of bioplastics. The FTIR spectra of the flours displayed a similar pattern, as seen in the profiles plotted in Figure 2A.

All samples showed a prominent peak at 3200–3600 cm^−1^, which corresponds to the stretching vibration of hydroxyl (OH) groups, which could be present in the water content in the flour or in the terminal groups of protein chains. The intensity of the peak varied depending on the presence of hydroxyl groups in the sample, with the taro sample exhibiting a broader peak compared to the banana and yucca samples. This broader peak could indicate higher hydrophilicity and moisture-absorption properties in the resulting bioplastics [30]

The peaks detected at 2940 cm^−1^ in the spectroscopic analysis indicate the stretching vibration of methyl C-H bonds, suggesting the presence of methyl groups in the sample. Additionally, the peak observed at 1055 cm^−1^ corresponds to the stretching vibration of C=O bonds in primary alcohols, indicating the presence of primary alcohols with carbonyl functional groups in the sample [46].

The peaks observed at 1600–1750 cm^−1^ in the spectroscopic analysis correspond to the stretching vibrations of carbonyl (C=O) bonds, as well as OH vibrations. These peaks indicate the presence of residual bonded water molecules and suggest the potential presence of biopolymer candidates with favorable film-forming properties. Moreover, the samples exhibited some peaks at 900–1200 cm^−1^, which correspond to starch (C-O-C) stretching vibrations. This suggests the presence of starch-based biopolymers that could contribute to the film-forming and mechanical properties of the resulting bioplastics, as referenced in [47,48]

Based on the FTIR characterization, all three types of Honduran stabilized agro-wastes show potential as suitable candidates for the fabrication of bioplastics, each with their own unique advantages in terms of molecular composition and functional groups.

#### 3.2.3. Granulometry

Figure 2B shows the particle-size distribution of the taro, yucca, and banana flours. It can be observed that taro presents the most polydisperse particle distribution of the three samples, with the most uniform being that obtained for the banana flour. However, banana had the highest average particle size (21.8 µm for banana compared to 7.9 and 11.7 µm for taro and yucca, respectively). Different authors state that a uniform particle size less than 200 µm favors the formation of more stable bioplastics [49]. In this sense, all samples would meet the size criteria, but banana flour should produce more stable bioplastics by having a more homogeneous size distribution. Note that the peak of the distribution was always below 60 µm, as this was the size of the slit that was used when sieving. There is a tail above this since there are always “elongated” particles that can slip through this slit at its smallest part.

#### 3.2.4. Differential Scanning Calorimetry (DSC)

Heat-flow patterns from DSC measurements in a temperature range between 20 °C and 250 °C are shown in Figure 3A for the three flours. Two endothermic DSC peaks can be observed. These peaks correspond to changes in the aggregation state of proteins. The first peak is located at 60 °C and can be attributed to the physical ageing phenomenon, which is a general process that occurs over time in glassy or partially glassy polymers below their glass-transition temperature and is a manifestation of the non-equilibrium nature of the glassy state [50]. This effect has also been detected in protein systems such as pea protein [35]. The second peak appears at 140 °C for banana flour and at approximately 180 °C for the taro and yucca flours, which may be related to protein denaturation at high temperatures. The differences between systems were due to the different compositions of the raw materials.

#### 3.2.5. TGA

Figure 3C shows the thermogravimetric analysis of the different flours. As can be observed in the TGA profiles, the three systems had a first weight loss between 50 °C and 100 °C, which was due to the evaporation of the water content. This weight loss was in the same range as the moisture content observed in Table 2. A more pronounced loss was observed between 250 °C and 350 °C, which was related to the loss of organic matter. However, 100% of the weight was not lost because the samples contained traces of ash that were not volatile at the temperatures studied. Similar results have been found in previous studies [51].

### 3.3. Bioplastics Characterization

#### 3.3.1. Dynamic-Mechanical Tests

Flexural and tensile frequency-sweep tests (Figure 4A and Figure 4B, respectively) were carried out for the three bioplastics manufactured. According to the results obtained, all the systems presented higher values for the elastic modulus (E′) than for the viscous modulus (E″), highlighting the solid character of the samples. In addition, the samples presented a similar profile, with a slight increase in the values of the elastic and the viscous modulus with frequency in both flexural and tensile modes.

To establish a better comparison between the different systems, the values of the elastic modulus (E′) at 1 Hz and the loss tangent (E″/E′) were evaluated (Table 3). Comparing the different systems, banana bioplastics presented the highest E′ values in both flexural and tensile modes. These results were consistent with the characterization of the raw materials since banana flour presented the most homogeneous particle-size distribution. Greater homogeneity in the particle size of the flour allows bioplastics to be formed with greater ease and homogeneity in their structure. This could imply that their mechanical properties are better and, consequently, E′ increases. For the same reason, taro bioplastics presented the smallest moduli values. Nevertheless, all the samples presented a notably solid character, since the loss tangent was always lower than 0.4.

They all presented better tensile strength than flexural strength. In this sense, the values of the moduli obtained in the tensile tests were higher. This can determine the applications of use for these bioplastics, since a greater stress requirement can be supported by them. In previous studies, the dynamic-mechanical properties of bioplastics derived from different sources have been investigated. For example, a previous study reported the E′1B values for bioplastics made from corn starch to be around 0.5–0.7 Pa [52], which is comparable to our findings for taro (0.69 ± 0.05 Pa). However, our study demonstrated that yucca-based bioplastics have a higher E′1B value of 1.46 ± 0.08 Pa, indicating superior mechanical properties compared to corn-starch-based bioplastics.

#### 3.3.2. Tensile Static Tests

The bioplastics were also subjected to tensile static tests to evaluate their resistance up to rupture. The stress–strain profiles can be observed in Figure 5. The three systems showed a similar profile, consisting of an initial linear region, corresponding to elastic deformation, followed by a plastic-deformation region, which was characterized by a continuous decrease in the stress–strain slope going through a maximum value. Finally, each system broke at a different strain value.

The Young’s modulus and the maximum stress and strain at break of the systems were obtained to compare the different systems (Table 3). As can be seen, banana bioplastics presented the highest Young’s modulus and maximum stress, although strain at break was similar in all of the samples. Therefore, the banana bioplastics seemed to be the ones that supported the most stress, as was also verified in the dynamic tests. However, this greater resistance did not make them more deformable than the other bioplastics evaluated. Furthermore, in terms of tensile static analysis, a previous study [53] examined the tensile modulus (E) of bioplastics derived from Kraft paper and from corn starch, reporting values of 60 MPa and 90 MPa, respectively. In comparison, our study demonstrated that the samples from our study exhibited significantly higher tensile modulus, exceeding 99 MPa. Notably, the banana-based bioplastics showed the highest tensile modulus of 308.5 ± 13.9 MPa, indicating a superior mechanical strength compared to both Kraft-paper- and corn-starch-based bioplastics.

Our study’s significance is highlighted through these comparisons, showcasing the mechanical performance of bioplastics derived from specific agro-wastes (taro, yucca, and banana) and their potential as alternatives to conventional plastics. By examining the mechanical properties of these bioplastics, our findings contribute valuable insights to the existing body of knowledge, highlighting their suitability for various applications.

#### 3.3.3. Water-Uptake Capacity (*WUC*)

*WUC* values for the different bioplastics are shown in Figure 6. As can be observed, bioplastics produced from taro had the highest water-uptake capacity, whereas those obtained from yucca had the lowest values. This may be related to the difference in the hydroxyl groups present in the materials, as already observed in the FTIR profiles, where taro had a slightly higher percentage than yucca and banana. These hydroxyl groups have a higher hydrophilicity, so they can increase the bioplastic’s ability to absorb and retain water. This capacity will also condition the final application of the bioplastics since there are sectors where bioplastics must be impermeable (packaging) and others where this absorption is beneficial (hygienic sector).

## 4. Conclusions

This study provides a comprehensive analysis and evaluation of three different Honduran agro-wastes (taro, yucca, and banana) as raw materials for bioplastics. The study successfully produced stable flours with suitable characteristics after a process of drying and grinding the agro-wastes. The taro flour presented the highest protein content (around 4.7%) and banana flour showed the highest moisture content (around 2%). The extrusion and injection molding of the mixtures of the different flours with plasticizers resulted in the production of bioplastics with different characteristics. The study found that banana bioplastics had the best mechanical properties (with a Young’s modulus around 300 MPa), while taro bioplastics had the highest water-uptake capacity (200%). Overall, the research contribution of this study is significant as it presents an eco-friendly and sustainable alternative to traditional plastics and promotes the circular economy in the agro-industrial sector. The study’s findings suggest that agro-industrial wastes can be used as raw materials for bioplastics, offering a sustainable solution and addressing gaps in the research related to the overreliance on non-renewable resources and the lack of biodegradability in traditional plastics. The developed bioplastics possess inherent properties that make them ideal for various industries, including the food, cosmetics, biomedical, and agricultural sectors.

The limitations of this study include the use of only three types of agro-wastes from Honduras, which may not be representative of other regions and may have different properties. Additionally, this study did not consider the economic viability of large-scale production of bioplastics using agro-wastes. Future research can explore the use of other agro-wastes and optimize the processing conditions for large-scale production. The study also did not address the environmental impact of the bioplastics’ disposal and degradation, which is an important consideration in promoting sustainability.

## Figures and Tables

**Figure 1 polymers-15-02625-f001:**
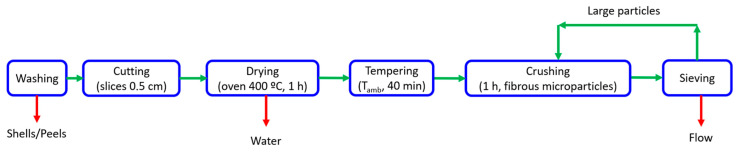
Flour preparation process.

**Figure 2 polymers-15-02625-f002:**
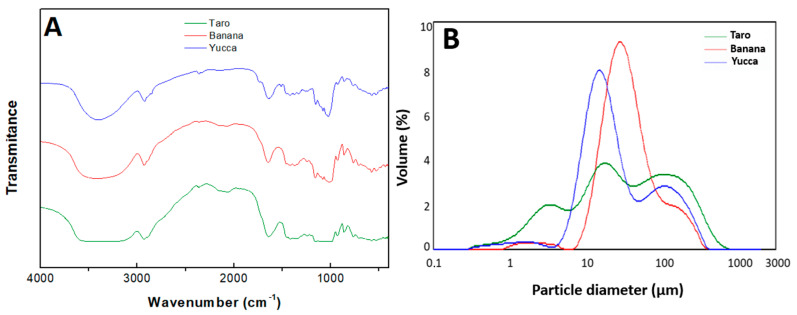
FTIR spectra of the three flours (yucca, banana, and taro) (**A**). Particle-size distribution of the three flours obtained (**B**).

**Figure 3 polymers-15-02625-f003:**
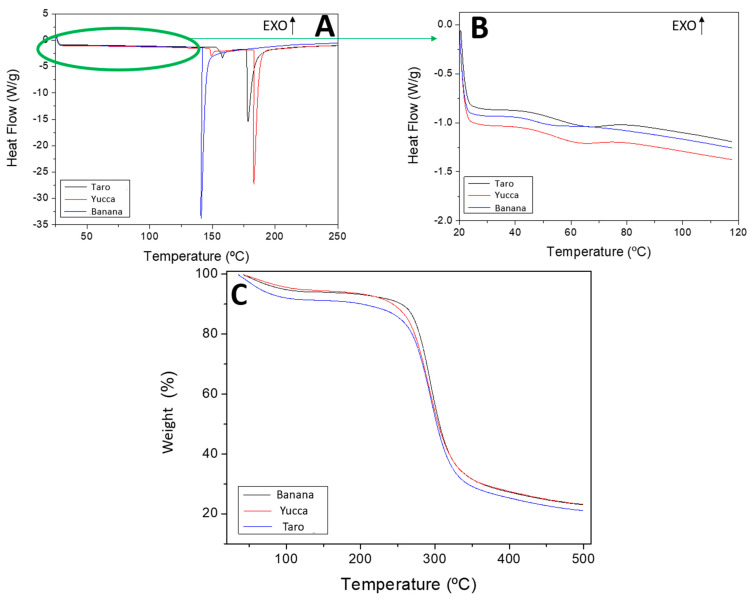
Thermal analysis of taro, yucca, and banana flours. (**A**) DSC profiles in a temperature range between 20 °C and 250 °C. (**B**) An amplification of DSC profiles. (**C**) TGA profiles in a temperature range between 50 °C and 500 °C.

**Figure 4 polymers-15-02625-f004:**
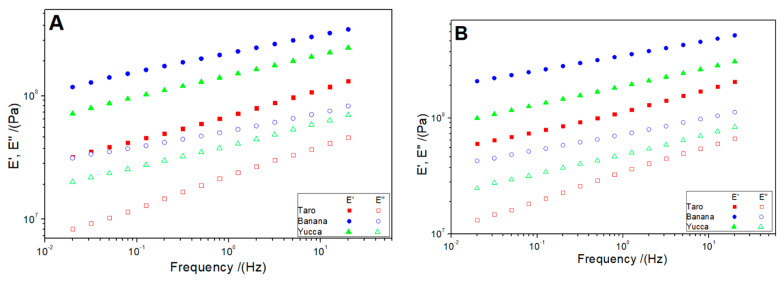
Flexural (**A**) and tensile (**B**) frequency sweeps for the bioplastics manufactured from taro, yucca, and banana.

**Figure 5 polymers-15-02625-f005:**
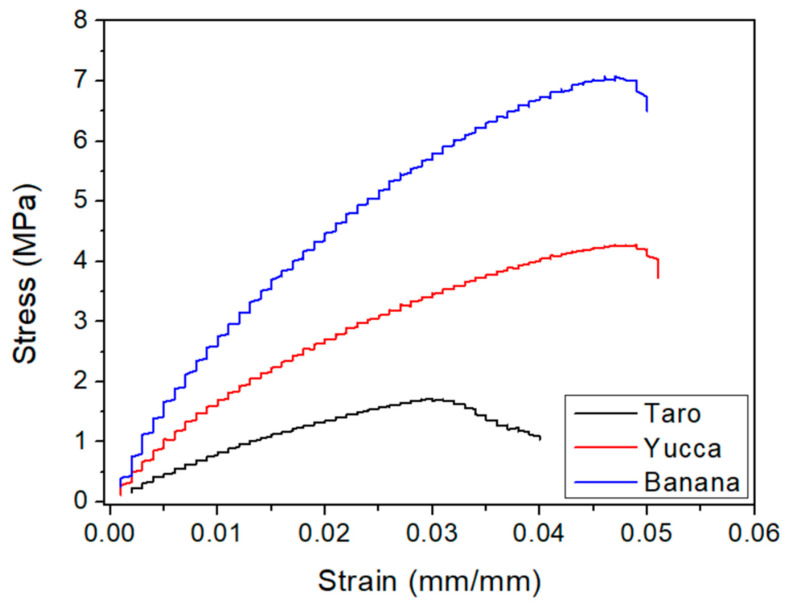
Stress–strain profiles of the different bioplastics.

**Figure 6 polymers-15-02625-f006:**
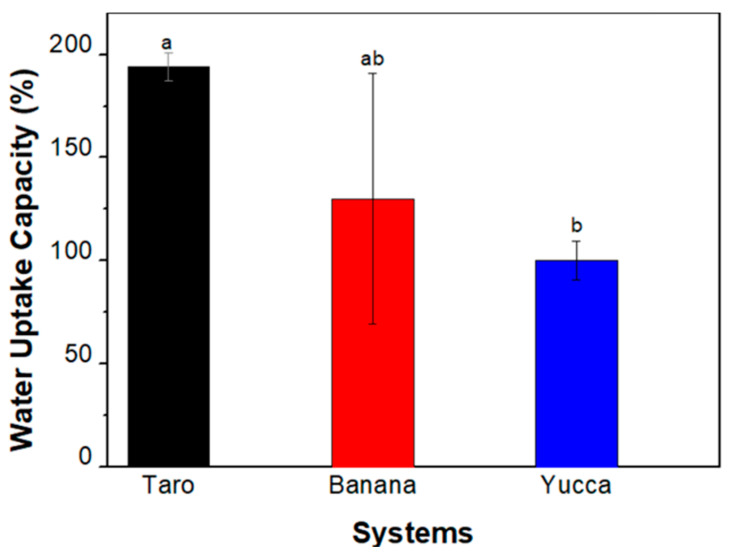
Water-uptake capacity for the different bioplastics. Different letters mean significant differences (*p* < 0.05).

**Table 1 polymers-15-02625-t001:** Yield of each step during flour obtention.

Sample	Peeling (%)	Cutting (%)	Drying (%)	Crushing + Sieving (%)
Taro	92	90	33	88
Yucca	80	98	37	92
Banana	56	99	45	93

**Table 2 polymers-15-02625-t002:** Chemical composition of the flour after stabilization of the three raw materials: taro, yucca, and banana.

Sample	Moisture (wt%)	Ashes (wt%)	Lipids (wt%)	Proteins (wt%)	Polysaccharides (wt%)
Taro	0.60 ± 0.04	6.90 ± 0.07	1.74 ± 0.61	4.71 ± 0.10	64.4 ± 1.3
Yucca	1.20 ± 0.03	5.45 ± 0.08	0.90 ± 0.02	2.87 ± 0.06	56.4 ± 3.4
Banana	1.50 ± 0.04	7.81 ± 0.05	0.22 ± 0.05	2.25 ± 0.17	59.2 ± 2.2

**Table 3 polymers-15-02625-t003:** Parameters of dynamic-mechanical tests: elastic modulus in flexural and tensile modes at 1 Hz (E′_1B_ and E′_1T_, respectively), loss tangent in flexural and tensile modes at 1 Hz (tan δ_1B_ and tan δ_1F_, respectively), and tensile static tests: Young’s modulus (E) and the maximum stress (σ_max_) and strain at break (ε_max_) for the bioplastics manufactured from taro, yucca, and banana.

System	Dynamic-Mechanical Tests				Tensile Static Analysis		
	Flexural Mode		Tensile Mode				
	E′_B_ (Pa)	tan δ_1B_ (-)	E′_1T_ (Pa)	tan δ_1F_ (-)	E (MPa)	σ_max_ (MPa)	ε_max_ (mm/mm)
Taro	0.69 ± 0.05	0.33 ± 0.02	1.13 ± 0.07	0.30 ± 0.04	99.5 ± 10.0	1.8 ± 0.4	0.03 ± 0.01
Yucca	1.46 ± 0.08	0.27 ± 0.03	1.89 ± 0.10	0.26 ± 0.02	187.6 ± 14.9	4.4 ± 0.1	0.05 ± 0.01
Banana	2.21 ± 0.11	0.24 ± 0.02	3.46 ± 0.15	0.22 ± 0.01	308.5 ± 13.9	6.3 ± 1.2	0.05 ± 0.01

## Data Availability

Data are available upon request to the authors.

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
