# Peer review of "Valorization of Honduran Agro-Food Waste to Produce Bioplastics"

_polymers, 2023, doi:10.3390/polym15122625_

Round 1

Reviewer 1 Report

1. Although this study has conducted a lot of analysis, there is no correspondence between the beginning and end, and there are no comments based on the research questions in the conclusions and recommendations
2. What is the question and purpose of this study? Please supplement
3. What are the motivations and gaps of this study?
4. What is the research contribution?
5. What are the research limitations and future recommendations?

This article looks more like a report, and research-oriented articles still need to be based on the corresponding research content and format

Moderate editing of the English language

Reviewer 2 Report

Castro-Criado and co-workers reported the process for the valorization of agri-food waste in Honduran into value-added bioplastics. The procedure was described very well and the yielded products were characterized by many techniques. The manuscript was written and organized very well, and the English usage was easy to understand. In summary, the research using local agro-residues as raw to produce bioplastics is interesting and I recommend a minor revision.

Some points that need to be revised:

The abstract seems like the introduction, and the necessary quantitative results and research data should be added in the abstract and the conclusion parts. 

Some previous works about update agri-food wastes to bioplastics should introduce and the previous reports about valorization of biomass wastes also should be cited to support the research background. 

The english usage is good.

Reviewer 3 Report

The manuscript reports the potential use of agro food waste for bioplastics production, there are several points that need to be improved/corrected before continue the publication process, following they are detailed:

- I recommend to correct the manuscript title, I think must be agro-food instead of agri-food.

-first time an abbreviation is written, must be defined, for instance PVC, among others.

- I recommend to correct the term "thermos-mechanical, the recommendation is to change by "thermo-mechanical" (line 44).

-Which are the main uses for Taro and Yucca? more details about this must be inserted in introduction section. 

-In experimental section, please indicate the sieve number used to obtain 60 micron of particle size.

-please correct the chemical fomulaes in the manuscript, I mean use super and under script numbers.

-Please indicate the material for ATR plate in FTIR, this, because in some situation the wavenumber range can vary depending of material.

-In section 2.2.5 indicate that TGA was carried out in a analyzer Q600, but this equipment of TA instruments correspond to a Simultaneous equipment (DSC/TGA), is this correct? I mean, it is not the same indicate a TGA equipment and a Simultaneous equipment.

-What kind of test were performed to establish the selected proportions for formulations in 70, 20 and 10%wt of flour, glycerol and water?

-In processing step indicate that a in-house extruder was used, but which is the screw geometry used?  it would be interesting if an image of this extruder is included as supplementary file. this because, considering that main component of used flours is starch, this kind of material has an interesting behavior when is extruded.

-Considering that starch main components are amylose and amylopectin, how the content of those is importan in final behavior of bioplastic? I mean, considering that polysaccharide content vary among materials between 7-8%wt, is it possible to identify the content of amylose and amylopectin in those agro-waste materials?

-in lines 251-252 indicate that there are diffrences in their molecular composition however according with FTIR spectra, this is not rigth, due all samples shows similar peaks.

-In line 260, indicate that a broader peak in FTIR spectra is associated to hydrophilicity and moisture absorption, but according with results reported in table 2 the taro does not has the higher moisture content. how can explain this?

-In Lines 299-300 indicate that content of proteins is different, according with endothermic peak around 140-180ºC, but according with data reported in table 2 protein content is quite similar for yuca and banana, but banana shows at lower temperature the denaturation process, please explain why.

-According with TGA thermogram, banana flour has a higher loss wieght associated to evaporation of water., how can relate this behavior with results reported in table 2?

-in lines 310-311, about the content of traces of ash, are there similar reports of this?

-can explain how the homogenity in particle size is associated with E`?

-About the Water uptake capacity, authors indicate that the difference can be associated with protein content in agro-waste, but according with data reported in table 2, the higher difference on this content is around 2%, is this difference a significant variation? I mean this slight variation can represent the difference in behavior of water uptake capacity? is this property possible to be explained with the presence of plasticizers?

-I recommend to authors to follow the instructions for authors, for references (not bibliography).

Round 2

Reviewer 1 Report

1. Still insufficient information on the background introduction
2. Lack of personal summary and reflection in the literature review

 Moderate editing of the English language

Reviewer 3 Report

After review the corrected version of manuscript, I wish to thanks to authors due this has been improved and most of observations/comments were taken in account, so based on that, my recommendation is ACCEPT AS IT
